# Priorities for Bark Anatomical Research: Study Venues and Open Questions

**DOI:** 10.3390/plants12101985

**Published:** 2023-05-15

**Authors:** Ilana Shtein, Jožica Gričar, Simcha Lev-Yadun, Alexei Oskolski, Marcelo R. Pace, Julieta A. Rosell, Alan Crivellaro

**Affiliations:** 1Department of Molecular Biology, Milken Campus, Ariel University, Ariel 40700, Israel; 2Eastern R&D Center, Milken Campus, Ariel 40700, Israel; 3Department of Forest Physiology and Genetics, Slovenian Forestry Institute, 1000 Ljubljana, Slovenia; 4Department of Biology & Environment, Faculty of Natural Sciences, University of Haifa-Oranim, Tivon 36006, Israel; 5Department of Botany and Plant Biotechnology, University of Johannesburg, Auckland Park, Johannesburg 2006, South Africa; 6Komarov Botanical Institute, Russian Academy of Science, Prof. Popov Str. 2, 197376 St. Petersburg, Russia; 7Instituto de Biología, Universidad Nacional Autónoma de México, Ciudad de México 04510, Mexico; 8Laboratorio Nacional de Ciencias de la Sostenibilidad, Instituto de Ecología, Universidad Nacional Autónoma de México, Ciudad de México 04510, Mexico; 9Forest Biometrics Laboratory, Faculty of Forestry, “Stefan cel Mare” University of Suceava, Str. Universitatii 13, 720229 Suceava, Romania

**Keywords:** bark, anatomy, cork, dilatation, periderm, phellogen, phenology, phloem, methods, ecology

## Abstract

The bark fulfils several essential functions in vascular plants and yields a wealth of raw materials, but the understanding of bark structure and function strongly lags behind our knowledge with respect to other plant tissues. The recent technological advances in sampling and preparation of barks for anatomical studies, along with the establishment of an agreed bark terminology, paved the way for more bark anatomical research. Whilst datasets reveal bark’s taxonomic and functional diversity in various ecosystems, a better understanding of the bark can advance the understanding of plants’ physiological and environmental challenges and solutions. We propose a set of priorities for understanding and further developing bark anatomical studies, including periderm structure in woody plants, phloem phenology, methods in bark anatomy research, bark functional ecology, relationships between bark macroscopic appearance, and its microscopic structure and discuss how to achieve these ambitious goals.

## 1. Bark Structure and Development

The bark (the cortex and epidermis in primary growth, and all the tissues outside the cambium in secondary growth), can be divided into inner and outer bark according to their developmental origin and status [1]. The cortex and epidermis that serve as primary bark is found only in young shoots, in thin roots, or in lineages that lack secondary growth (e.g., non-Asparagalean monocots). In shoots and roots that express cambial activity for a long time, the primary bark remains on the outer side, and it is commonly shed when the trunks become very thick. While the cortex and epidermis (all tissues outside the vascular cylinder) is produced by two primary apical meristems (ground meristem and protoderm, respectively) [1], the secondary bark is always the product of the activity of two secondary lateral meristems—the vascular cambium and the phellogen [1] (Figure 1). In many species, including herbaceous ones, another secondary process follows the increase in girth, causing cracks in the bark, a phenomenon known as dilatation. It may occur via cell expansion, random divisions, or via the activity of a dilatation meristem [2] (Figure 1B) (see below). The bark often includes cortex remnants, and consists of the secondary phloem, and a periderm or, in many cases, sequent periderms (rhytidome) and the tissues embedded in between them. All tissues found outwards the innermost periderm are considered as rhytidome [1]. Therefore, tissues such as the cortex and old phloem can within time be encapsulated by the periderms, and become part of the rhytidome.

The outermost protective layer of the bark is the periderm (Figure 1C). The periderm, formed by the lateral secondary meristem known as the phellogen (or in its common name, the cork cambium), is composed of dead phellem cells outwards, that are typically suberized and can be recognized by histological staining of suberin, and of phelloderm (live parenchymatic cells formed inwards by the phellogen), in addition to lenticels (ventilation shafts) [3]. Suberin is a hydrophobic lipid polymer, with varying chemical monomeric composition between species [4]. In *Quercus suber* (cork oak), the major model plant for cork structure, chemistry, and development, suberin accounts for about 40% of the phellem cell wall components, lignin being the second most abundant constituent (~25%) [5]. In comparison, Betula pendula’s outer bark was composed of 45% suberin, and only 9% lignin [6], while in *Calotropis procera*’s cork there is only 5% suberin and 40% lignin [7]. Cork cell wall composition varies greatly between individual trees [5] and there are many unanswered questions regarding eco-physiological significance of this variation between species, ecotypes and individuals. We stress that not all cork cells are suberized. Considering the lenticels, they are structurally variable. They are commonly composed of alternating bands of hard and soft layers. Their central part is composed of loosely arranged cells that allow ventilation. The cells can be suberized or not [3]. The cork of various species, e.g., *Pinus halepensis*, is composed of alternating layers of phellem cells, and very thick-walled lignified sclereids (stone cells) (Lev-Yadun, unpublished).

Bark thickness increases with organ maturation, while bark structure also shows axial changes [8,9,10,11,12]. The commonly enormous differences in bark structure and morphology in young versus mature organs (Figure 4C) reflects a combination of functional solutions, developmental constraints, and evolutionary history. There are still many open questions concerning these age- and ecology-related differences.

Although there are several important studies of bark development [8,10,13,14,15], the description of structural and morphological bark ontogeny is still partial for most species. It should be done along with documenting genetic and environmental related variability. Moreover, mechanical and biotic stresses cause significant changes in bark structure and chemistry [16]. Since practically all plants suffer some damages with time, the description of any species’ bark should include traumatic bark.

## 2. Bark Dilatation

When plant stems or roots grow in diameter, the bark can expand to a certain, but limited, degree. To avoid the formation of bark cracks, a special tissue is formed in the outer bark—a dilatation tissue [17] (Figure 1B and Figure 4B). Dilatation is the outcome of one or two processes that co-occur in many species. The first process is dilatation growth as the result of cell expansion or random cell divisions. Certain parenchyma cells of the primary cortex, of the primary and especially secondary phloem axial and ray parenchyma expand or divide to keep the bark intact. In other cases, groups of these cells enter a phase of cell division and form a dilatation meristem (Figure 1B), which has been recorded in many distinct, unrelated taxa, such as *Tilia* spp. (Malvaceae) and Cordia (Boraginaceae) [18,19]. The dilatation meristem, when formed, is typically not a continuous one such as the vascular cambium, but rather many separate small bands of dilatation meristems. When dilatation occurs in the phloem rays, a distinct funnel-shaped phloem ray dilatation occurs, as seen in transverse sections [2]. The dilatation meristem can be formed in the center of the ray or in the ray’s outer layer [2]. Dilatation may be regular or very irregular, resulting in certain cases in the formation of whirled tissues and various modified shapes of rays. Application of ethylene, wounding, or environmental conditions that induce ethylene production, such as flooding, result in higher dilatation activity [20].

## 3. Hormonal Control of Bark Structure

A basic understanding of the hormonal regulation of bark and cork formation emerges from their cellular composition and patterns of development. The fact that the secondary bark is produced in concert by the cambium, the phellogen, and by dilatation- all three regulated to a certain extent by ethylene—can attest to overlapping hormonal regulation of the activity of these meristems. Moreover, the fact that we know a lot about the hormonal regulation of differentiation of some of the cell types that compose the bark (both primary and secondary) adds to our understanding of the hormonal control of bark development. For instance, auxin is the major regulator of phloem element differentiation and does it in a polar way [21]. A combination of mostly auxin and gibberellin regulates fiber formation [22].

Bark morphology can also illuminate the hormonal regulation of bark formation, especially of the cork. Cork as a rule is not formed in fast growing shoots and it has been proposed that the lack of cork in such organs allows for better photosynthesis [23]. The above is suggests that auxin suppresses cork formation, a possibility further supported by the fact that cork initiation is delayed around buds and below leaves [24]. These holes and cracks in the cork were found to allow for quicker canopy regeneration in cases of canopy breakage [24]. Like the rays and periderm, dilatation is also positively controlled by ethylene [20].

The combination of morphological, anatomical, and developmental bark responses to environmental factors, especially sun irradiation has allowed to propose the mode of hormonal regulation of bark structure. Some of the anatomical evidence (e.g., [25,26]) was known a long time before the role of plant hormones in bark differentiation was considered. Refs. [25,26] found a strong correlation between ray structure and ray size, with the type of lenticels (an integral part of the cork system) in stems of a very long list of woody dicotyledon species. He showed that the orientation of lenticels- transverse versus axial (longitudinal)- depends on the type of rays. Refs. [25,26] showed that in woody plants with aggregate rays made of the union of many unicellular rays, and in which the compounding process occurs only after many years, as well as in plants with regular short rays, there are transverse lenticels. Axial lenticels are associated with tall rays, aggregate rays that have multiseriate subunits, and with aggregate rays that have uniseriate subunits that complete their aggregation within the first three years. The hormone suggested to be the main inducer of cork formation is ethylene [9]. Ethylene was also found to be the inducer of ray formation [27]. Thus, the fact that [25,26] found a strong correlation between ray structure, ray size, and the type of lenticels seems to fit well into the suggestion that ethylene regulates the differentiation of both structures.

Bark has many defensive functions, and not surprisingly, its development is also influenced not only by ethylene [9,20], but also by the stress/defense hormone methyl jasmonate (MJ) [28,29]. MJ was found to induce wounding-like responses in Pinaceae bark, including accumulation of phenolic compounds, cell swelling and traumatic duct formation in phloem—though there was no induced periderm formation [30]. Those MJ responses in conifers were further shown to be mediated by ethylene [29].

While plant hormones are the upstream regulators of bark formation and cell differentiation, they do not do it directly, but via changing the expression and activity of a very long list of genes including regulatory ones. Though a good coverage of this issue is outside the scope of this review; we give below a few references to this subject [31,32,33].

## 4. Architectural Types of Bark: Relationships between Macroscopic Appearance and Anatomical Structure

The tremendous diversity of bark macrostructure and its anatomical background still needs to be explored. The relations between macro- and microstructure of bark have been considered by few authors [34,35,36,37,38,39,40]. Although [41,42] made significant efforts to standardize the terms for macroscopic bark traits, the further development of descriptive terminology for bark appearance and its integration with anatomical concepts are still required [17].

To a large extent, external bark appearance is determined by two functional features: (1) by the ability or disability of the outermost layers of bark to maintain its continuity in the course of tangential expansion (dilatation, see above), and (2) by the presence or absence of separation layers, i.e., the layers of fragile tissues enabling a regular abscission of outer portions of bark. Diverse barks can be classified into four major architectural types: stretched, exfoliating, furrowed and peeling barks (Figure 2), and combinations of these.

Stretched (e.g., *Adansonia za*, Figure 2A) and exfoliating barks (e.g., *Platanus* sp., Figure 2B) share the ability of conspicuous expansion without superficial disruptions, usually with the occurrence of lenticels and eye-like markings on their surface. While the stretched barks have a smooth appearance, the exfoliating ones can be recognized by scars of fallen scales. The barks of both types share a thin periderm or rhytidome underlain by a continuous parenchymatous layer (pseudocortex or proliferation tissue *sensu* [36] or/and with sectors of thin-walled cells formed by dilatation meristems [39]. The exfoliating barks are distinctive from the stretching ones by the presence of superficial separation layers made of thin-walled phellem cells [40].

Unlike the barks of these two types, the furrowed (e.g., *Quercus suber,* Figure 2C) and peeling (e.g., *Buddleja saligna*, Figure 2D) barks share prominent axial fissures on their surfaces in response to the dilatation, usually without lenticels and eye-like markings. The furrowed barks lack separation layers; they are mostly covered by a conspicuous persistent rhytidome without an underlying parenchymatous layer [43]. The peeling barks are distinctive from the furrowed ones by prominent detachment of their outer portions. Such barks share relatively thin rhytidomes. Their periderms may serve as separation layers, whereas the nonconducting phloem (including the collapsed layers when found) that remains after the periderm is shed performs a protective function [44]. In the peeling bark of some species (e.g., *Vitis vinifera*, *Ulmus glabra*), however, these functions are allocated differently: the periderm is responsible for protection, while the nonconducting phloem—for the separation of abscised portions [10,45].

The shifts between the peeling and furrowed bark types can be associated with age variation- for instance, in many pine species. Further research is required to explore the anatomical and functional features of these bark types in more detail and to clarify their ecological and evolutionary implications.

## 5. Phloem

The inner bark of older trees consists of secondary phloem derived from the vascular cambium and composed of conducting and nonconducting phloem (Figure 1). The sieve elements of the phloem are usually conductive for only one or two growing seasons; however, the nonconducting phloem remains functional for several years in many ways, such as storing and mobilizing starch and other metabolites and maintaining the meristematic capacity to form phellogen or dilatation tissue [17]. The growing interest and recent advances in quantitative studies of bark/phloem anatomy, combined with novel analytical tools, are opening new opportunities to study the interactions between radial growth and climate of trees in different environments [46,47,48,49,50]. Trees originating in different climates have been shown to have different cambial rhythms and cell differentiation processes, which are also reflected in phloem structure (Figure 3) [51]. Thus, environmental signals stored in phloem structure may contain complementary information to that stored in xylem traits, allowing better assessment of tree performance and prediction of adaptive capacity and vulnerability of trees to climate change, including stress events [52,53,54,55].

Unlike xylem, however, recent studies show that seasonal cambial rhythm and age-related changes in phloem, such as the collapse of conducting elements, have a major influence on the width and structure of its conducting part. For example, in the temperate tree species *Picea abies*, *Fagus sylvatica* and *Quercus petraea*, the proportion of phloem increment formed in the previous growing season is highest at the beginning of the growing season, accounting for more than 80% of the conducting phloem. In the second part of the growing season, the proportion of phloem increment formed in the previous year gradually decreases and reaches a minimum at the end of the growing season presenting less than 40% of the conducting phloem in *Picea abies* and *Quercus petraea* and about 20% in *Fagus sylvatica*. The relative proportions of early phloem to late phloem change seasonally, which also affects the phloem potential conducting efficiency due to morphological differences in the conduits [52]. Moreover, the fine structure of phloem conduits, such as the characteristics of sieve plates and sieve plate pores, can also have a major influence on phloem conducting potential [17,56].

The width of the conducting phloem is species-specific, being on average 30% and 55% wider in *Quercus petraea* than in *Picea abies* and *Fagus sylvatica*, respectively. However, in addition to the seasonal changes in structure, the width of the conducting phloem also varies during the growing season. It is generally narrowest in spring at the beginning and widest at the end of cambial cell production and is affected by two opposing processes: (a) the formation of new phloem cells by the cambium (positive contribution) and (b) the collapse of older phloem cells (negative contribution). Seasonal differences in width and structure of the conducting phloem indicate that the progression of these two processes is not synchronized [51]. While the cambium of temperate tree species undergoes a dormant phase to survive harsh winter conditions [57,58], the collapse process appears to continue in winter. This explains the narrowest conducting phloem in spring, as no new phloem cells are produced during winter to compensate for the collapse process. Since not much is known about the temporal dynamics of age-related processes in older bark tissues and on their dependence on various internal and environmental factors, this topic needs to be investigated in future studies. Besides the effect of seasonality, the structure and width of the conducting phloem depend also on genetic factors; consequently, variability exists between individuals of the same tree species as well [59].

The above examples illustrate the need to consider the timing of sampling to ensure reliable data interpretation when comparing results from different laboratories on an intra-annual time scale [51]. Hence, in addition to published list of microscopic features for bark identification [17], the establishment of a common sampling protocol and methodology is necessary to properly evaluate the conducting potential of phloem sieve elements in different tree species [51].

## 6. Bark Ecology

Although our understanding of the functional ecology of bark is still limited, especially in comparison with wood, it has significantly improved in the last decade. One of the main reasons for this has been a shift from studies aiming to explain the remarkably high variation in bark traits based solely on protection against fire, to studies that recognize that bark is shaped by diverse functions [60,61,62]. Certainly, bark protects stems from fire and other external hazards [63], a function served mainly by its outer region [64]. However, bark also transports photosynthates [65], stores water, carbohydrates, and many other compounds [66,67,68,69], has a mechanical role in stem support especially in young and thin stems and branches [70,71], and can carry out photosynthesis [72]. As in any multifunctional structure, trade-offs evolved between bark functions. For example, a thicker outer corky bark will impede the penetration of light for photosynthesis but will provide better protection to stems [73]. Understanding bark ecology thus requires considering issues beyond a fire-centered perspective and taking into account its different functions and selective drivers to understand bark’s remarkable diversity.

In general, we know very little about the function of bark colors. There are many tree species with brownish or grayish barks, but there are many other species in which the bark is conspicuously colorful (Figure 4A). It can be so when the outer bark of the whole trunk is of one color, or made of patches of different colors. It has been proposed that such coloration can serve to undermine herbivorous insect camouflage, signal toxicity (aposematism), and in white deciduous trees of the temperate and boreal regions to camouflage the trees from mammalian herbivores during winter [74,75]. It has also been suggested that the white bark cools the cambium during winter in order to avoid thawing and thus prevents damage during repeated freezing [76], but this function seem to be relevant in relatively moderate temperate latitudes rather than in tropical ones [74]. We also lack a good understanding of the importance of non-color-based defense from herbivory adaptations that are common in barks, including prickles, thorns, toxins, and mechanical hardness.

Although significant progress has been made in understanding how different functions play out in bark’s structural diversity, we are still far from fully understanding bark’s structure-function relationships. This is for sure one of the most promising venues for future research in bark functional ecology. For example, it is unclear which are the tissues or main regions that drive variation in bark density or bark storage capacity. A bark with high density tends to be thinner, has lower storage capacity, and also lower decomposition rates [77,78]. Moreover, the density of bark, especially of its inner living region, i.e., inner to the innermost periderm, covaries strongly with sapwood density [67,79], suggesting that these two stem regions are functionally and developmentally coordinated. This coordination is still in need of further research, as well as the role of rays in the exchange of water and other molecules across the cambium [80,81]. Also, while there are some accepted functional implications of certain bark morphologies such as that smooth barks cause insect herbivores to fall off them [82], the functional role of most morphologies is still unclear. Documenting how anatomical and morphological variation causes functional differences in bark would help to disentangle the selective factors that lead to the very high diversity of barks in the woody plant world.

## 7. Overcoming the Methodological Limitation

The main reason repeatedly reported for why barks are much less studied than all others plant tissues and organs is the methodological limitation [14,17]. Barks combine hard, lignified and also suberized cells, with soft, non-lignified ones, and sectioning them commonly result in the tearing of the entire structure, or detaching of the bark from the wood. If dried, all the non-lignified cells may collapse, and obtaining good sections gets even harder. Yet, across the years, methods have been developed to overcome these limitations. In Figure 5, all these procedures are summarized in a flowchart and their detailed protocols can be found in [83]. Here we will focus on why each step is recommended.

The first step is always fixing the collected bark samples in liquid. Fixatives such as paraformaldehyde or FAA (10% formaldehyde, 5% acetic acid, 70% ethyl alcohol [84]) can be used depending on the size of the sample (see flowchart for deciding), but in the absence of one of them, even common ethyl alcohol is better than drying. The sooner the material is fixed, the better, for the phloem tends to rapidly collapse. If a vacuum chamber is available, it is highly recommended, since it forces the fixative inside the sample yielding better fixation. After a couple of days, the material can be transferred to 70% ethanol and conserved for as long as necessary. Herbaceous and soft barks do not need softening, but barks that combine sclerenchyma with non-lignified cells are better sectioned after softening, which can be done by boiling in water with some glycerin (10 mL/liter), which is a gentler method, or with the use of softening chemicals, such as ethylenediamine (ETD) or hydrofluoric acid. ETD is preferred, since hydrofluoric acid is extremely hazardous [85]. Usually a couple of days in 10% ETD is sufficient for making the tissue more manageable for sectioning [86], and the solution can be reused many times, even if getting very dark [85]. Embedding the samples is another step that greatly improves the quality of the sections. If embedding in paraffin or glycol methacrylate, a rotary microtome will be used for sectioning, while if using polyethylene glycol (PEG 1500), a sliding microtome will be used instead. The advantages of PEG 1500 are that it is water soluble and does not make lignified samples harder. PEG 1500 has in addition excellent penetration, and large samples can be embedded, even entire branches [87,88]. If working in a very humid environment, the addition of a 10% PEG 4000 on the last stage of the process helps making the block less prone to absorbing room humidity and easier to handle. To perform the sections, it is recommendable to use of a polystyrene resin made of Styrofoam dissolved in either xylene or butyl acetate, which is applied upon the sample as a coat that holds the section together, avoiding tearing while facilitating sectioning, staining and dehydrating [89].

The study of bark anatomy requires a staining method that singles out lignified from non-lignified tissues. This is fundamental to aid in the correct identification of the different cell types that compose the bark, and to understand bark development and function (e.g., biomechanics). Barks with a lot of fibers, for instance, almost entirely lack sieve tube collapse, even in their non-conducting parts [90]. Double-staining can be done by different methods, such as Astra Blue and Safranin, Fast Green and Safranin or Resorcine Blue and Ferrid Chloride. Another options is the use of a metachromatic staining (Toluidine Blue O), which will stain differently lignified and unlignified tissues even though being a single staining. German authors have created a highly efficient method that combines Astra blue and Safranin in 50% ethanol creating a solution named Safrablau, which double stains with just some drops in 5–15 min [91]. The recipe consists of dissolving 1g of Astra blue in 100 mL of 50% ethanol and 1g of safranine dissolved in 50% ethanol, and then making a mix of 9 parts of Astra blue to 1 part of safranine (e.g., 90 mL of astra blue mixed to 10 mL of safranine). This percentage can be changed for species where the lignified tissues stain more hardly, to 7:3 or 8:2 [91]. Sometimes bleaching before staining (20% bleach in water) can help obtaining even better staining results, but the bleach has to be completely rinsed out before applying the staining, otherwise the stains precipitate. The presence of suberin, a lipidic substance common in phellem cells, can be done by the use of any of the Sudan stains (Sudan III, IV or Sudan Black) [17]. Once stained, permanent slides can be mounted by rinsing the sections in increasingly more concentrated solutions of ethanol up to absolute alcohol, a final rinse in either butyl acetate or xylene and mounted in either a natural (e.g., Canada balsam) or synthetic mounting material. The slides can be analyzed after the resin dries.

## 8. Concluding Remarks

Bark structure is scarcely studied compared to wood or leaf anatomy. However, this is a broad topic with many unanswered questions and a crucial one to understand the remarkably wide variation in bark morphology and function. The idea for this article arrived after a Q-net (Quantitative Wood Anatomy Network, https://qwa-net.com/(accessed on 30 April 2023)) symposium on bark structure that the authors of this article organized and participated in. We realized that there is a lot of interest in this topic in the scientific community, while at the same time there is a general lack of knowledge and methodological know-how. This article tries to provide an overview of bark structural and functional anatomy, as well as to propose venues for future research. Many open questions yet remain; some were discussed throughout the article, and several are mentioned below.

External appearance of bark depends largely on its ability or disability to maintain its continuity in the course of tangential expansion, and the presence or absence of separation layers enabling a regular abscission of its outer portions. Four major types of bark architecture can be distinguished on this basis. Anatomical background of bark stretching without superficial disruption as well as the allocation of protective and separation layers within outer bark in different plant groups are, however, still poorly known.

Similarly, the ecological significance of bark features such as color, photosynthetic function, density, and more is poorly understood. Moreover, the link between bark structure and environmental stresses is yet mostly unexplored.

Bark biomechanics is a largely unexplored topic. Is dilatation induced by mechanic tension or is it only a hormonal response? This topic merits an interdisciplinary study with precise measurements.

Phloem research lags behind xylem largely because of technical issues. How does the phloem structure–function relationship change under stress? In order to answer this question, sampling methodology must be clarified.

We hope that this short review will stimulate interest in bark research and encourage new collaborations and studies of this important and largely ignored topic.

## Figures and Tables

**Figure 1 plants-12-01985-f001:**
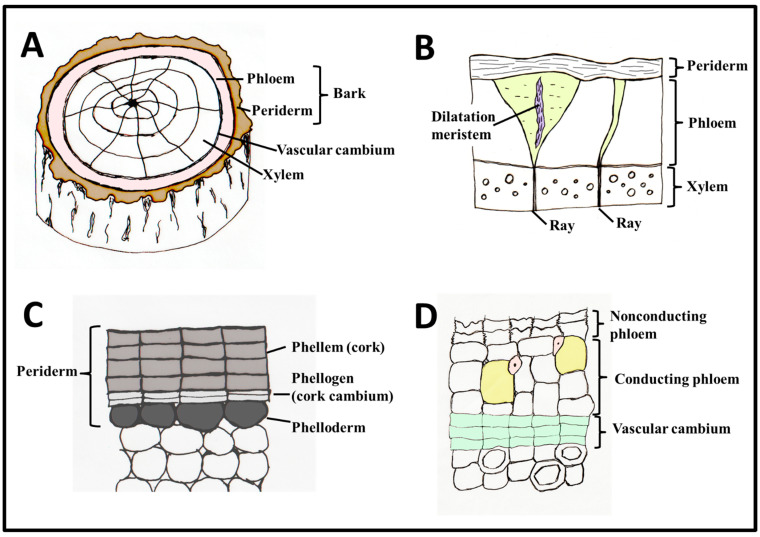
Features of secondary bark anatomy. (**A**) Secondary bark occurs in woody plant parts and comprises secondary phloem and periderm. Sometimes, primary bark (cortex) can persist between secondary phloem and periderm (not shown). (**B**) When the stem matures, dilatation tissues can be produced, and sometimes, dilatation meristems arise in phloem rays. (**C**) The periderm is produced by a meristem—the phellogen, or cork cambium. (**D**) The secondary phloem is produced by the vascular cambium and is comprised of younger conducting phloem and older nonconducting phloem, which is often collapsed. Also see Glossary. Illustration drawn by I.S.

**Figure 2 plants-12-01985-f002:**
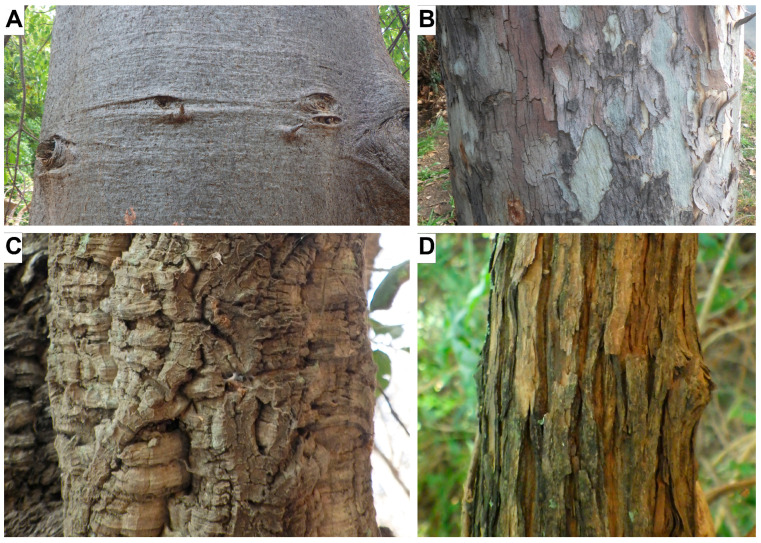
Macroscopic appearance of major architectural types of bark: (**A**) Stretched bark of *Adansonia za*; (**B**) Exfoliating bark of *Platanus* sp.; (**C**) Furrowed bark of *Quercus suber*; (**D**) Peeling bark of *Buddleja saligna*.

**Figure 3 plants-12-01985-f003:**
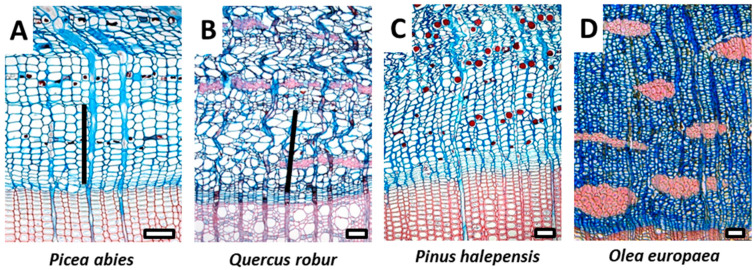
Different structure of conducting phloem growth rings in temperate vs. Mediterranean species. Visible phloem growth rings (black line) in temperate tree species *Picea abies* (**A**) and *Quercus robur* (**B**), while in the Mediterranean species *Pinus halepensis* (**C**) and *Olea europaea* (**D**), the demarcation of the youngest growth rings is not feasible. Scale bars = 100 µm.

**Figure 4 plants-12-01985-f004:**
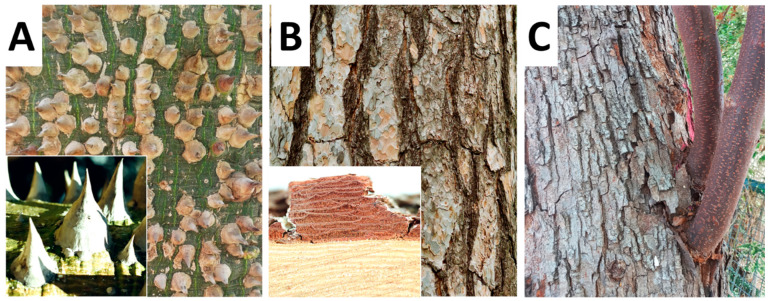
Various morphological features of tree bark. (**A**) *Ceiba insignis*—green bark with prickles, (**B**) *Pinus sylvestris*—bark rhytidome, (**C**) *Koelreuteria bipinnata*—different bark types on the same plant.

**Figure 5 plants-12-01985-f005:**
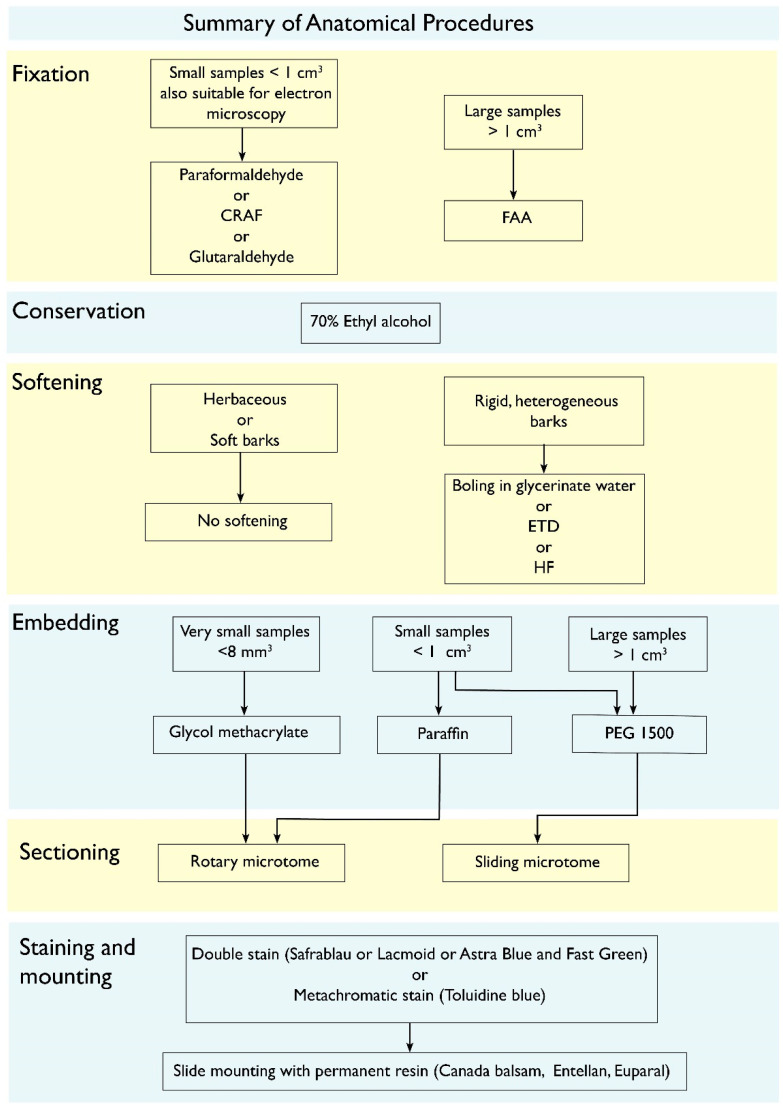
Flowchart of the technical procedure for bark histology. After Pace (2019). For details, see text.

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
