# Peer review of "Priorities for Bark Anatomical Research: Study Venues and Open Questions"

_plants, 2023, doi:10.3390/plants12101985_

Round 1

Reviewer 1 Report

Dear authors,

the manuscript “Priorities for bark anatomical research: study venues and open questions” provides a very good overview on bark structural and functional anatomy and gives important input for future research. However, some anatomical descriptions might be confusing for the readership regarding their terminology and the information in the last chapter should be better combined with the former chapters (why is the staining done?). Please find my general and specific remarks below.

General remarks

A. A list of terminologies will be helpful for the reader since several terms (for the same tissue) are used in the different chapters such as primary and secondary barks, outer bark, rhytidome, “cork”, primary cortex and some more.

B. There should be also information about suberin staining in the last chapter. The present descriptions are very useful for investigation of the primary bark, secondary phloem and phelloderm but not necessarily for phellem cells.

Specific remarks

1. Page 1, first chapter, line 7: primary bark: the synonym “cortex” might be provided

2. Page 1, first chapter, line 10: (and also in chapter 2. “Bark dilatation”): please provide as well classical anatomy textbook citations (e.g. Esau 1964, Evert 2006 (Esaus plant anatomy), Angyalossy et al. 2016 (IAWA list of microscopic bark features)). Esau 1964 (Structure and Development of the Bark in Dicotyledons. In: Zimmermann MH (ed.). The Formation of Wood in Forest Trees. Academic Press, New York, London: 37-50) writes “This increase may occur through repeated orderly divisions in the median part of the ray so that one might speak of a dilatation meristem (Schneider, 1955)”. Thus, only if ray cell divisions are restricted in the median region of wedge-shaped rays a so-called “dilatation meristem” is present and can be termed as such (Angyalossy et al. 2016). In many cases, cell expansion and anticlinal cell divisions occur throughout multiseriate rays. All this becomes clearer with the figures (even though the Figure might post the message that such a dilatation meristem is quite often present, and not restricted to only some species….) but it should be also explained more detailed in the text (here or in the chapter 2. Bark dilatation).

3. Page 1, first chapter, line 11: the last sentence is confusing; do you mean “inflation” of cells?

4. Page 2, lines 2-3: the terminology “rhytidome” is positioned at the wrong place, since the rhytidome includes both the older sequent periderms and the tissues embedded between them. A list of all bark features described in your review would be helpful. In addition, a citation is missing here.

5. Page 2, lines 4-6: phellem should not be termed as cork, maybe as “cork”, since, in many species, not all cells of the phellem are suberized. See e.g. the review of Trockenbrodt (1990)!

6. Page 2, line 10: please explain the terminology “outer bark”, it is not clear what is meant here – simply the phellem or already the rhytidome. In any case, a clarification that the phellem can include non-suberized cells as well, would be helpful.

Page 6: Figure 1C would delete “cork” here, also “cork cambium” is a rather antique terminology

Page 6, Legend of Figure 1: mention that between the secondary bark and periderm there is also the primary bark (cortex) – please be consistent with the terminology – otherwise readers not familiar with bark anatomy will get lost in the text

Page 3, line 6: this expansion is called “inflation”?

Page 4, line 14: This = There?

Page 4, line 22: which evidence?

Page 4, line 26: terming lenticels as a part of the cork system is oversimplified; you may refer to a recent review about lenticels: Rosner and Morris (2023) (DOI 10.1163/22941932-bja10090)

Page 4, lines 28-30: please rephrase this sentence; the message is difficult to catch

Page 4, line 35: repeating, see line 14

Page 4, line 42: delete “anatomy”

Page 4, line 49: it = is

Page 5, line 13: Fig. 3 = Fig.2

Page 5, line 17: the origin of such structures might be defined, overgrown branches etc.

Page 5, line 20, phelloderm produced by the phellogen?

Page 7, lines 7-8: you may mention that this is also dependent on the genetics and thus variability exists between individuals of the same species, there are differences between fast- and slow-growing individuals; and slow growing individuals use more increment of the previous increment for conducting sugars – see e.g. Rosner et al. (2001) (DOI 10.1007/s00468-001-0131-9)

Page 9, lines 33-34: why is it important to separate lignified from non-lignified tissues? Please give some information why and how the results can be useful to solve the research questions raised in the former chapters (e.g. regarding astrablue/safranine especially for secondary phloem research and the biomechanics of the bark). Regarding the phellem it is also very important to indicate the suberized structures (see your chapter 1 about suberin) which can be done easily with Sudan stains (Angyalossy et al. 2016: Sudan III, IV, or Sudan Black B). For a practical example please see also the review of Rosner and Morris (2023) (DOI 10.1163/22941932-bja10090). The authors provide a staining method where non-lignified, lignified and suberized cell walls of phellem tissues can be identified all together on a given bark section.

Page 9, line 45: it is not relevant that the authors are from Germany, especially when the literature is not available in English but in Portuguese language. Therefore, could you please provide the recipe?

Page 11, line 18: delete one “this”

Page 11: good conclusion!

Author Response

General remarks

  1. A list of terminologies will be helpful for the reader since several terms (for the same tissue) are used in the different chapters such as primary and secondary barks, outer bark, rhytidome, “cork”, primary cortex and some more.

A:

Thank you for your suggestion. We added a glossary:

“Glossary of terms (after Angyalossy et al. 2016), also see Fig.1:

Bark - all tissues outside the vascular cambium. Can be composed of inner and outer bark.

Outer bark - includes periderm and rhytidome.

Periderm – Secondary protective tissue that replaces the epidermis. Consists of phellem (cork), phellogen (cork cambium), and phelloderm.

Phellogen (or cork cambium) - a lateral meristem forming the periderm.

Phellem (cork) – Part of the periderm, formed by the phellogen (cork cambium). Protective tissue composed of non-living suberized cells.

Secondary phloem – phloem tissue derived from the vascular cambium, composed of conducting and nonconducting phloem.

Dilatation - increase in the circumference of the bark by parenchyma cell division and cell expansion in order to adjust to the secondary growth of the stem.

Cortex - primary ground tissue region between the vascular system and the dermal tissue.

Lenticels - isolated region in the periderm distinguished from the phellem, which has intercellular spaces and apparently participates in gas exchange.”

  1. There should be also information about suberin staining in the last chapter. The present descriptions are very useful for investigation of the primary bark, secondary phloem and phelloderm but not necessarily for phellem cells.

 A:

Included.

Specific remarks

  1. Page 1, first chapter, line 7: primary bark: the synonym “cortex” might be provided

A:

We changed the text to “While the cortex (all tissues outside the vascular cylinder) is produced by two primary apical meristems (protoderm and ground meristem) (Fahn 1990), the secondary bark is always the product of the activity of two secondary lateral meristems - the vascular cambium and the phellogen (Fahn 1990)”

  1. Page 1, first chapter, line 10: (and also in chapter 2. “Bark dilatation”): please provide as well classical anatomy textbook citations (e.g. Esau 1964, Evert 2006 (Esaus plant anatomy), Angyalossy et al. 2016 (IAWA list of microscopic bark features)). Esau 1964 (Structure and Development of the Bark in Dicotyledons. In: Zimmermann MH (ed.).

A:

We added a citation of Fahn, as an excellent classical anatomy text.

The Formation of Wood in Forest Trees. Academic Press, New York, London: 37-50) writes “This increase may occur through repeated orderly divisions in the median part of the ray so that one might speak of a dilatation meristem (Schneider, 1955)”. Thus, only if ray cell divisions are restricted in the median region of wedge-shaped rays a so-called “dilatation meristem” is present and can be termed as such (Angyalossy et al. 2016). In many cases, cell expansion and anticlinal cell divisions occur throughout multiseriate rays. All this becomes clearer with the figures (even though the Figure might post the message that such a dilatation meristem is quite often present, and not restricted to only some species….) but it should be also explained more detailed in the text (here or in the chapter 2. Bark dilatation).

A:

Thank you for drawing our attention to this. We added the following : “The dilatation meristem can be formed in the center of the ray or in the ray’s outer layer (Lev-Yadun 1996).” One of the co-authors (SLY) worked extensively on dilatation, and it indeed can occur not only in the center of the ray.

  1. Page 1, first chapter, line 11: the last sentence is confusing; do you mean “inflation” of cells?

A:

We expanded the text to include the following: “In many species including herbaceous ones, another secondary process following the increase in girth that may cause cracks in the bark occurs and known as dilatation. It may occur via cell expansion or via the activity of a dilatation meristem (Lev-Yadun 1996) (Fig. 1B) (see below).

  1. Page 2, lines 2-3: the terminology “rhytidome” is positioned at the wrong place, since the rhytidome includes both the older sequent periderms and the tissues embedded between them.

A:

We elaborated: “All tissues found outwards the innermost rhytidome are considered as rhytidome (Fahn 1990). Therefore, tissues such as the cortex and old phloem can with time become parts of the rhytidome.”

A list of all bark features described in your review would be helpful. In addition, a citation is missing here.

A:

We added a glossary.

  1. Page 2, lines 4-6: phellem should not be termed as cork, maybe as “cork”, since, in many species, not all cells of the phellem are suberized. See e.g. the review of Trockenbrodt (1990)!

A:

Thank you. We elaborated: “We stress that not all cork cells are suberized. Considering the lenticels, they are structurally variable. They are commonly composed of alternating bands of hard and soft layers. Their central part is composed of loosely arranged cells that allow ventilation. The cells can be suberized or not (Rosner & Morris 2022). The cork of various species, e.g., Pinus halepensis, is composed of alternating layers of phellem cells, and very thick-walled lignified sclereids (stone cells) (Lev-Yadun, unpublished).”

  1. Page 2, line 10: please explain the terminology “outer bark”, it is not clear what is meant here – simply the phellem or already the rhytidome. In any case, a clarification that the phellem can include non-suberized cells as well, would be helpful.

A:

We changed the text to “The bark (all tissues outside the primary xylem and phloem in primary growth, or outside the cambium in secondary growth), can be divided into inner and outer barks according to their developmental origin and status (Fahn 1990). The cortex that serves as a primary bark is found only in young shoots, in thin roots, or in lineages that lack secondary growth (e.g., most monocots). In shoots and roots that express cambial activity for a long time, the primary bark remains on the outer side, and it is commonly shed when the trunks become very thick. While the cortex (all tissues outside the vascular cylinder) is produced by two primary apical meristems (protoderm and ground meristem) (Fahn 1990), the secondary bark is always the product of the activity of two secondary lateral meristems - the vascular cambium and the phellogen (Fahn 1990) (Fig. 1). In many species including herbaceous ones, another secondary process following the increase in girth that may cause cracks in the bark occurs and known as dilatation. It may occur via cell expansion or via the activity of a dilatation meristem (Lev-Yadun 1996) (Fig. 1B) (see below). The bark often includes cortex remnants, and mainly consists of the secondary phloem, and in many cases also sequent periderms (rhytidome) and the tissues embedded in between them. All tissues found outwards the innermost rhytidome are considered as rhytidome (Fahn 1990). Therefore, tissues such as the cortex and old phloem can with time become parts of the rhytidome.”

Page 6: Figure 1C would delete “cork” here, also “cork cambium” is a rather antique terminology

A:

“Cork cambium” is a commonly used terminology. We changed to “The outermost protective layer of the bark is the periderm, or cork tissue (Fig. 1C). The periderm, formed by the lateral secondary meristem known as the phellogen (or in its common name, the cork cambium), is composed of dead phellem cells, that are typically suberized.”

Page 6, Legend of Figure 1: mention that between the secondary bark and periderm there is also the primary bark (cortex) – please be consistent with the terminology – otherwise readers not familiar with bark anatomy will get lost in the text

A:

We added to the legend “Sometimes primary bark (cortex) can persist between secondary phloem and periderm (not shown).”

Page 3, line 6: this expansion is called “inflation”?

A:

No. We meant “expansion”.

Page 4, line 14: This = There?

A:

We changed to “The above”

Page 4, line 22: which evidence?

A:

We elaborated “Some of the anatomical evidence”

Page 4, line 26: terming lenticels as a part of the cork system is oversimplified; you may refer to a recent review about lenticels: Rosner and Morris (2023) (DOI 10.1163/22941932-bja10090)

A:

Rosner and Morris excellent article shows that lenticels are a part of the periderm. Those are indeed interesting structures with distinct function, but it’s out of the scope of this review. We changed the text to “lenticels (an integral part of the cork system)”. We added elsewhere in the text “Considering the lenticels, they are structurally variable. They are commonly composed of alternating bands of hard and soft layers. Their central part is composed of loosely arranged cells that allow ventilation. The cells can be suberized or not (Rosner & Morris 2022).”

Page 4, lines 28-30: please rephrase this sentence; the message is difficult to catch

A:

We rephrased to : “He showed that the orientation of lenticels, transverse versus axial (longitudinal), depends on the type of rays. Wetmore (1926a, 1926b) showed that in woody plants with aggregate rays composed of the union of unicellular rays, plants in which the compounding process of rays occurs only after many years, and plants with regular short rays, all have transverse lenticels.”

Page 4, line 35: repeating, see line 14

A:

We removed the redundant part.

Page 4, line 42: delete “anatomy”

A:

Done

Page 4, line 49: it = is

A:

Done

Page 5, line 13: Fig. 3 = Fig.2

A:

Done

Page 5, line 17: the origin of such structures might be defined, overgrown branches etc.

A:

The separation layers can be of different origin (layers of thin-walled phellem cells, entire phelloderm, or non-conducting secondary phloem), as it is explained in two following paragraphs.

Page 5, line 20, phelloderm produced by the phellogen?

A:

Yes. Also see glossary.

Page 7, lines 7-8: you may mention that this is also dependent on the genetics and thus variability exists between individuals of the same species, there are differences between fast- and slow-growing individuals; and slow growing individuals use more increment of the previous increment for conducting sugars – see e.g. Rosner et al. (2001) (DOI 10.1007/s00468-001-0131-9)

A:

We thank the reviewer for the suggestion. We added a sentence: “Besides the effect of seasonality, the structure and width of the conducting phloem depend also on genetic factors; consequently, variability exists between individuals of the same tree species as well (e.g., Rosner et al. 2001).”

Page 9, lines 33-34: why is it important to separate lignified from non-lignified tissues? Please give some information why and how the results can be useful to solve the research questions raised in the former chapters (e.g. regarding astrablue/safranine especially for secondary phloem research and the biomechanics of the bark).

Regarding the phellem it is also very important to indicate the suberized structures (see your chapter 1 about suberin) which can be done easily with Sudan stains (Angyalossy et al. 2016: Sudan III, IV, or Sudan Black B). For a practical example please see also the review of Rosner and Morris (2023) (DOI 10.1163/22941932-bja10090). The authors provide a staining method where non-lignified, lignified and suberized cell walls of phellem tissues can be identified all together on a given bark section.

A:

Included.

Page 9, line 45: it is not relevant that the authors are from Germany, especially when the literature is not available in English but in Portuguese language. Therefore, could you please provide the recipe?

A:

Included.

Page 11, line 18: delete one “this”

A:

Done

Page 11: good conclusion!

A:

Thank you ?

Reviewer 2 Report

The manuscript entitled “Priorities for bark anatomical research: study venues and open questions” presents a review about bark fulfils several essential functions in vascular plants and yields a wealth of raw materials, but understanding bark structure and function strongly lag behind our knowledge with respect to other plant tissues.

After careful consideration and review the manuscript, I found that the results presented in your paper are not appropriate for publication. The authors have focused their review on very superficial aspects of Bark structure and development, Bark dilatation, Hormonal control of bark structure, Architectural types of bark, Phloem and Bark ecology, with a very basic approximation. One of the key features of bark is its multi-layered structure. The outermost layer is called the periderm, which is made up of dead cells that protect the underlying living tissue. Beneath the periderm is the cortex, which contains living cells responsible for transporting nutrients and water from the roots to the leaves. The innermost layer of bark is called the phloem, which is responsible for transporting the sugars produced by photosynthesis from the leaves to the rest of the tree. Anatomical research on bark has also revealed the presence of specialized cells and tissues that contribute to its function. For example, some trees have lenticels, which are small pores in the bark that allow for gas exchange between the inner and outer layers. Other trees have resin ducts, which produce resin to deter pests and pathogens. Bark can also contain sclereids, which are specialized cells that provide structural support and protection against herbivores. Research has also shown that bark can vary greatly in its composition and structure across different tree species. For example, some trees have smooth bark, while others have rough, textured bark. Some species have thick, corky bark, while others have thin, flaky bark. These variations can have important implications for a tree's survival, as different bark types can offer different levels of protection against environmental stresses.

Here are some potential priorities for understanding and further developing bark anatomical studies and that should be considered in a review:

-Investigating the function and evolution of specialized bark tissues, such as resin ducts and sclereids, in different tree species.

-Examining how bark structure and composition vary in response to environmental stresses such as drought, heat, and disease.

-Studying the role of bark in carbon storage and sequestration, as well as its potential for use in sustainable materials and bioenergy.

-Developing new techniques and technologies for studying bark anatomy, such as advanced microscopy and imaging tools.

-Exploring the relationships between bark anatomy and other aspects of tree biology, such as growth, reproduction, and symbiotic interactions with other organisms.

-Investigating the ecological and evolutionary consequences of bark anatomy, such as how it affects tree defense against herbivores and pathogens, or how it contributes to the diversification of tree lineages.

-Collaborating across disciplines, including botany, ecology, biotechnology, and materials science, to explore the full potential of bark anatomy research in addressing pressing environmental challenges.

Therefore, I feel that the research presented in your manuscript does not contribute significantly to the existing body of knowledge in the field and I must reject your publication.

Finally, I would like to request that you make some improvements to the figures in your manuscript before publication. While your figures are generally clear and informative, I believe that some changes could help to better convey your results to readers.

Author Response

After careful consideration and review the manuscript, I found that the results presented in your paper are not appropriate for publication. The authors have focused their review on very superficial aspects of Bark structure and development, Bark dilatation, Hormonal control of bark structure, Architectural types of bark, Phloem and Bark ecology, with a very basic approximation. One of the key features of bark is its multi-layered structure. The outermost layer is called the periderm, which is made up of dead cells that protect the underlying living tissue. Beneath the periderm is the cortex, which contains living cells responsible for transporting nutrients and water from the roots to the leaves. The innermost layer of bark is called the phloem, which is responsible for transporting the sugars produced by photosynthesis from the leaves to the rest of the tree. Anatomical research on bark has also revealed the presence of specialized cells and tissues that contribute to its function. For example, some trees have lenticels, which are small pores in the bark that allow for gas exchange between the inner and outer layers. Other trees have resin ducts, which produce resin to deter pests and pathogens. Bark can also contain sclereids, which are specialized cells that provide structural support and protection against herbivores. Research has also shown that bark can vary greatly in its composition and structure across different tree species. For example, some trees have smooth bark, while others have rough, textured bark. Some species have thick, corky bark, while others have thin, flaky bark. These variations can have important implications for a tree's survival, as different bark types can offer different levels of protection against environmental stresses.

A:

We thank the reviewer for the thorough reading of our manuscript. Indeed bark is a complex tissue with many anatomical features. Several reviews could be written on this subject. We chose to focus on several topics and the rest were out of our scope.

Here are some potential priorities for understanding and further developing bark anatomical studies and that should be considered in a review:

-Investigating the function and evolution of specialized bark tissues, such as resin ducts and sclereids, in different tree species.

-Examining how bark structure and composition vary in response to environmental stresses such as drought, heat, and disease.

-Studying the role of bark in carbon storage and sequestration, as well as its potential for use in sustainable materials and bioenergy.

-Developing new techniques and technologies for studying bark anatomy, such as advanced microscopy and imaging tools.

-Exploring the relationships between bark anatomy and other aspects of tree biology, such as growth, reproduction, and symbiotic interactions with other organisms.

-Investigating the ecological and evolutionary consequences of bark anatomy, such as how it affects tree defense against herbivores and pathogens, or how it contributes to the diversification of tree lineages.

-Collaborating across disciplines, including botany, ecology, biotechnology, and materials science, to explore the full potential of bark anatomy research in addressing pressing environmental challenges.

A:

Again, we thank the reviewer for pointing out the potential of bark anatomy and the priorities that should be addressed. We agree with the suggestions. Many of the proposed topics are reviewed in our paper. As we wrote in the concluding remarks, there are major gaps in bark research, and many topics are not studied yet. However, in this opinion paper, based on our expertise, we have focused on topics related to periderm structure in woody plants, phloem phenology, methods of bark anatomy research, bark functional ecology, and relationships between the bark macroscopic appearance and its microscopic structure. We hope that in the future as much research as possible will be done on this topic that will attempt to answer the questions posed.

Therefore, I feel that the research presented in your manuscript does not contribute significantly to the existing body of knowledge in the field and I must reject your publication.

Finally, I would like to request that you make some improvements to the figures in your manuscript before publication. While your figures are generally clear and informative, I believe that some changes could help to better convey your results to readers.

Reviewer 3 Report

Dear authors,

Thank you for an interesting and partly very well written paper.

I have a few suggestions/concerns after my review.

The structure of the article is a mix of state of the art, own studies, discussion, research proposal and methodology. This leads to some confusions. I suggest you to be more clear about what is what, also regarding your own findings/contributions. A traditional review article with a concluding remark on future research would be better.

Also, I am not really used to the way references are handled. At several places they seem to be missing, as in figure 1? In other places it is not clear what text is taken from what reference. For example, the second last paragraph in 3. If the reference to the first sentence is Hudgins, it should be stated already there. The same “problem” is seen in many sections, making it unclear. (see also the beginning of 2 and 5, and the second section on pg 7.) It could be that I come from another discipline of research, this is not correct as far as I am concerned. Also, connected to this, I am not used to the use of “we” and “us” or the way the concluding remarks is written. Is appears to be more like a research proposal than a paper?

Yours

Author Response

Also, I am not really used to the way references are handled. At several places they seem to be missing, as in figure 1?

A:

Figure 1 is original drawing made by one of the authors- as stated in the figure legend.

In other places it is not clear what text is taken from what reference. For example, the second last paragraph in 3. If the reference to the first sentence is Hudgins, it should be stated already there.

A:

The first sentence only states the general conclusions, it’s without reference. The next two sentences are from 2 different studies- referenced- and they give examples to the general statement. All three are written in one single paragraph separate from other topics, so in our opinion it’s not confusing.

The same “problem” is seen in many sections, making it unclear. (see also the beginning of 2 and 5, and the second section on pg 7.)

It could be that I come from another discipline of research, this is not correct as far as I am concerned.

A:

Thank you. The authors of this review are experts in the field of bark research, and many statements are simply part of their common knowledge and experience.  We added some more references.

Also, connected to this, I am not used to the use of “we” and “us” or the way the concluding remarks is written. Is appears to be more like a research proposal than a paper?

A:

It’s true that traditionally it was customary to use third person in articles and official texts. However, the use of first rather than third person pronouns is becoming very common, and it’s actually advised to used first person as it engages the reader.

Round 2

Reviewer 2 Report

The authors have done a great job revising the manuscript, improving it substantially, and although bark is a complex tissue with many anatomical features, the presented paper has improved in style and background. I recommend its publication after the revisions made.

Author Response

We thank the reviewer for the kind words and the reviewing effort.

Reviewer 3 Report

Thank you for your answers and updated version.

I do not agree with you that the following sentence is general:

Bark has many defensive functions, and not surprisingly, its development is also influenced not only by ethylene, but also by the stress/defense hormone methyl jasmonate (MJ)

If it is from own research, state that. Otherwise, reference should be in the first sentence of the section. This is what I teach my students. (Also valid for other places in the text). I guess it is up to the Editor to decide the format of the text.

(remove the “1” from the first paragraph.)

Author Response

Thank you for your answers and updated version.

 We thank the reviewer for many useful comments.

I do not agree with you that the following sentence is general:

 Bark has many defensive functions, and not surprisingly, its development is also influenced not only by ethylene, but also by the stress/defense hormone methyl jasmonate (MJ)

 If it is from own research, state that. Otherwise, reference should be in the first sentence of the section. This is what I teach my students. (Also valid for other places in the text). I guess it is up to the Editor to decide the format of the text.

 Thank you. We added several references.

(remove the “1” from the first paragraph.)

Done.
